# DISCRETE SEQUENTIAL PREDICTION OF CONTINUOUS ACTIONS FOR DEEP RL

## ABSTRACT

It has long been assumed that high dimensional continuous control problems cannot be solved effectively by discretizing individual dimensions of the action space due to the exponentially large number of bins over which policies would have to be learned. In this paper, we draw inspiration from the recent success of sequence-to-sequence models for structured prediction problems to develop policies over discretized spaces. Central to this method is the realization that complex functions over high dimensional spaces can be modeled by neural networks that predict one dimension at a time. Specifically, we show how Q-values and policies over continuous spaces can be modeled using a next step prediction model over discretized dimensions. With this parameterization, it is possible to both leverage the compositional structure of action spaces during learning, as well as compute maxima over action spaces (approximately). On a simple example task we demonstrate empirically that our method can perform global search, which effectively gets around the local optimization issues that plague DDPG. We apply the technique to off-policy (Q-learning) methods and show that our method can achieve the state-of-the-art for off-policy methods on several continuous control tasks.

## 1 INTRODUCTION

Reinforcement learning has long been considered as a general framework applicable to a broad range of problems. However, the approaches used to tackle discrete and continuous action spaces have been fundamentally different. In discrete domains, algorithms such as Q-learning leverage backups through Bellman equations and dynamic programming to solve problems effectively. These strategies have led to the use of deep neural networks to learn policies and value functions that can achieve superhuman accuracy in several games (Mnih et al., 2013; Silver et al., 2016) where actions lie in discrete domains. This success spurred the development of RL techniques that use deep neural networks for continuous control problems (Lillicrap et al., 2015; Gu et al., 2016a; Levine et al., 2016). The gains in these domains, however, have not been as outsized as they have been for discrete action domains.

This disparity is, in part, a result of the inherent difficulty in maximizing an arbitrary function on a continuous domain, even in low-dimensional settings. Furthermore, it becomes harder to apply dynamic programming methods to back up value function estimates from successor states to parent states in continuous control problems. Several of the recent continuous control reinforcement learning approaches attempt to borrow characteristics from discrete problems by proposing models that allow maximization and backups more easily (Gu et al., 2016a).

One way in which continuous control can avail itself of the above advantages is to discretize each of the dimensions of continuous control action spaces. As noted in (Lillicrap et al., 2015), doing this naively, however, would create an exponentially large discrete space of actions. For example with $M$ dimensions being discretized into $N$ bins, the problem would balloon to a discrete space with $M^N$ possible actions.

We leverage the recent success of sequence-to-sequence type models (Sutskever et al., 2014a) to train such discretized models, without falling into the trap of requiring an exponentially large number of actions. Our method relies on a technique that was first introduced in (Bengio & Bengio, 1999), which allows us to escape the curse of dimensionality in high dimensional spaces by modeling complicated

probability distributions using the chain rule decomposition. In this paper, we similarly parameterize functions of interest – Q-values – using a decomposition of the joint function into a sequence of conditional values tied together with the bellman operator. With this formulation, we are able to achieve fine-grained discretization of individual domains, without an explosion in the number of parameters; at the same time we can model arbitrarily complex distributions while maintaining the ability to perform (approximate) global maximization. These benefits come at the cost of shifting the exponentially complex action space into an exponentially complex MDP (Bertsekas et al., 1995; De Farias & Van Roy, 2004). In many settings, however, there are relationships between transitions that can be leveraged and large regions of good solutions, which means that this exponential space need not be fully explored. Existing work using neural networks to perform approximate exponential search is evidence of this Vinyals et al. (2015); Bello et al. (2016).

While this strategy can be applied to most function approximation settings in RL, we focus on off-policy settings with an algorithm akin to DQN. Empirical results on an illustrative multimodal problem demonstrates how our model is able to perform global maximization, avoiding the exploration problems faced by algorithms like NAF (Gu et al., 2016b) and DDPG (Lillicrap et al., 2015). We also show the effectiveness of our method on a range of benchmark continuous control problems from hopper to humanoid.

## 2 METHOD

In this paper, we introduce the idea of building continuous control algorithms utilizing sequential, or autoregressive, models that predict over action spaces one dimension at a time. Here, we use discrete distributions over each dimension (achieved by discretizing each continuous dimension into bins) and apply it using off-policy learning.

### 2.1 PRELIMINARIES

We briefly describe the notation we use in this paper. Let $\boldsymbol{s}_t \in \mathbb{R}^L$ be the observed state of the agent, $\boldsymbol{a} \in \mathbb{R}^N$ be the $N$ dimensional action space, and $\mathcal{E}$ be the stochastic environment in which the agent operates. Finally, let $\boldsymbol{a}^{i:j} = \begin{bmatrix} a^i \cdots a^j \end{bmatrix}^T$ be the vector obtained by taking the sub-range/slice of a vector $\boldsymbol{a} = \begin{bmatrix} a^1 \cdots a^N \end{bmatrix}^T$.

At each step $t$, the agent takes an action $\boldsymbol{a}_t$, receives a reward $r_t$ from the environment and transitions stochastically to a new state $\boldsymbol{s}_{t+1}$ according to (possibly unknown) dynamics $p_{\mathcal{E}}(\boldsymbol{s}_{t+1}|\boldsymbol{s}_t, \boldsymbol{a}_t)$. An episode consists of a sequence of such steps $(\boldsymbol{s}_t, \boldsymbol{a}_t, r_t, \boldsymbol{s}_{t+1})$, with $t = 1 \cdots H$ where $H$ is the last time step. An episode terminates when a stopping criterion $F(\boldsymbol{s}_{t+1})$ is true (for example when a game is lost, or when the number of steps is greater than some threshold length $H_{max}$).

Let $R_t = \sum_{i=t}^{H} \gamma^{i-1} r_i$ be the discounted reward received by the agent starting at step $t$ of an episode. As with standard reinforcement learning, the goal of our agent is to learn a policy $\pi(\boldsymbol{s}_t)$ that maximizes the expected future reward $\mathbb{E}[R_H]$ it would receive from the environment by following this policy.

Because this paper is focused on off-policy learning with Q-Learning (Watkins & Dayan, 1992), we will provide a brief description of the algorithm.

#### 2.1.1 Q-LEARNING

Q-learning is an off-policy algorithm that learns an action-value function $Q(\boldsymbol{s}, \boldsymbol{a})$ and a corresponding greedy-policy, $\pi^Q(\mathbf{s}) = \operatorname{argmax}_{\boldsymbol{a}} Q(\boldsymbol{s}, \boldsymbol{a})$. The model is trained by finding the fixed point of the Bellman operator, i.e.

$$Q(\boldsymbol{s_t}, \boldsymbol{a_t}) = \mathbb{E}_{\boldsymbol{s_{t+1}} \sim p_{\mathcal{E}}(\cdot|\boldsymbol{s_t}, \boldsymbol{a_t})}[r + \gamma Q(\boldsymbol{s_{t+1}}, \pi^Q(\boldsymbol{s_{t+1}}))] \qquad \forall(\boldsymbol{s_t}, \boldsymbol{a_t}) \qquad (1)$$

This is done by minimizing the Bellman Error, over the exploration distribution, $\rho_{\beta}(\boldsymbol{s})$

$$L = \mathbb{E}_{\boldsymbol{s_t} \sim \rho_{\beta}(\cdot), \boldsymbol{s_{t+1}} \sim \rho_{\mathcal{E}}(\cdot|\boldsymbol{s_t}, \boldsymbol{a_t})} \|Q(\boldsymbol{s_t}, \boldsymbol{a_t}) - (r + \gamma Q(\boldsymbol{s_{t+1}}, \pi^Q(\boldsymbol{s_{t+1}})))\|^2 \qquad (2)$$

Traditionally, $Q$ is represented as a table of state action pairs or with linear function approximators or shallow neural networks (Watkins & Dayan, 1992; Tesauro, 1995). Recently, there has been an

effort to apply these techniques to more complex domains using non-linear function approximators that are deep neural networks (Mnih et al., 2013; 2015). In these models, a *Deep Q-Network* (DQN) parameterized by parameters, $\theta$, is used to predict Q-values, i.e. $Q(\boldsymbol{s}, \boldsymbol{a}) = f(\boldsymbol{s}, \boldsymbol{a}; \theta)$. The DQN parameters, $\theta$, are trained by performing gradient descent on the error in equation 2, without taking a gradient through the Q-values of the successor states (although, see (Baird, 1995) for an approach that takes this into account).

Since the greedy policy, $\pi^Q(\boldsymbol{s})$, uses the action value with the maximum Q-value, it is essential that any parametric form of $Q$ be able to find a maxima easily with respect to actions. For a DQN where the output layer predicts the Q-values for each of the discrete outputs, it is easy to find this max – it is simply the action corresponding to the index of the output with the highest estimated Q-value. In continuous action problems, it can be tricky to formulate a parametric form of the Q-value where it is easy to find such a maxima. Existing techniques either use a restrictive functional form, such as NAF (Gu et al., 2016b). DDPG (Lillicrap et al., 2015) employs a second neural network to approximate this max, in addition to the $Q$ function approximator. This second network is trained to maximize / ascend the $Q$ function as follows: $\text{J} = \text{E}_{s \sim \rho_\beta}[Q(s, \mu(a; \theta^\mu); \theta^Q)]$
$$\nabla_{\theta^\mu} J = \mathbb{E}_{s \sim \rho_\beta}[Q(s, \mu(a; \theta^\mu); \theta^Q) \nabla_{\theta^\mu} \mu(s; \theta^\mu)],$$

where $\rho_\beta$ is the state distribution explored by some behavioral policy, $\beta$ and $\mu(\cdot; \theta^\mu)$ is the deterministic policy.

In this work we modify the form of our Q-value function while still retaining the ability to find local maxima over actions for use in a greedy policy.

## 2.2 SEQUENTIAL DQN

In this section, we outline our proposed model, Sequential DQN (SDQN). This model decomposes the original MDP model with $N$-D actions to a similar MDP which contains sequences of 1-D actions. By doing this, we have 2 layer hierarchy of MDP – the "upper" containing the original environment, and the "lower" containing the transformed, stretched out, MDP. Both MDP model the same environment. We then combine these MDP by noting equality of $Q$ values at certain states and doing bellman backups against this equality. See figure 1 for a pictorial view of this hierarchy.

Consider an environment with states $\boldsymbol{s_t}$ and actions $\boldsymbol{a} \in \mathbb{R}^N$. We can perform a transformation to this environment into a similar environment replacing each $N$-D action into a sequence of $N$ 1-D actions. This introduces a new MDP consisting of states $\boldsymbol{u}_k^{s_t}$ where superscript denotes alignment to the state $\boldsymbol{s_t}$, above, and subscript $k$ to denote time offset on the lower MDP from $\boldsymbol{s_t}$. As a result, $\boldsymbol{u}_k^{s_t} = (\boldsymbol{s_t}, \boldsymbol{a}^{1:k})$ is a tuple containing the state $\boldsymbol{s_t}$ from original MDP and a history of additional states in the new MDP – in our case, a concatenation of actions $\boldsymbol{a}^{1:k}$ previously selected. The transitions of this new MDP can be defined by two rules: when all 1-D actions are taken we compute 1 step in the $N$-D environment receiving a new state, $\boldsymbol{s_{t+1}}$, a reward $r_t$, and resetting $\mathbf{a}$. In all other transitions, we append the previously selected action in $\mathbf{a}$ and receive 0 reward.

This transformation reduces the $N$-D actions to a series of 1-D actions. We can now discretize the 1-D output space and directly apply $Q$-learning. Note that we could apply this strategy to continuous values, without discretization, by choosing a conditional distribution, such as a mixture of 1-D Gaussians, over which a maxima can easily be found. As such, this approach is equally applicable to pure continuous domains as compared to discrete approximations.

The downside to this transformation is that it increases the number of steps needed to solve the transformed MDP. In practice, this transformation makes learning a $Q$-function considerably harder. The extra steps of dynamic programming coupled with learned function approximators causes large overestimation and stability issues. This can be avoided by learning $Q$-values for both MDPs at the same time and performing the bellman backup from the lower to the upper MDP for the transitions, $s_t$, where $Q$-values should be equal.

We define $Q^U(\boldsymbol{s}, \boldsymbol{a}) \in \mathbb{R}$, $\boldsymbol{a} \in \mathbb{R}^N$, as a function that is the $Q$-value for the top MDP. Next, we define $Q^L(\boldsymbol{u}, a^i) \in \mathbb{R}$ where $a^i \in \mathbb{R}$ as the $Q$ value for the lower MDP. We would like to have consistent $Q$-values across both MDPs when they perform one step in the environment. To make this possible,

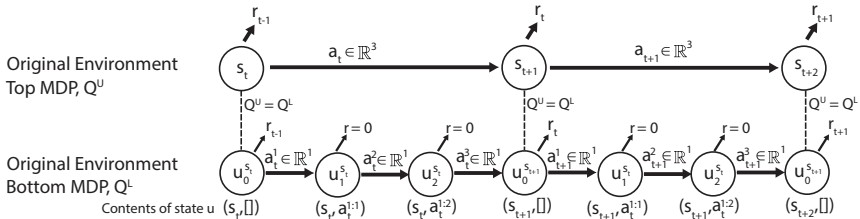

Figure 1: Demonstration of a transformed environment with three dimensional action space. New states, $\boldsymbol{u}$ are introduced to keep the action dimension at each transition one dimensional. The values of these states are shown bellow the circles. Each circle represents a state in the MDP. The transformed environment's replicated states are now augmented with the previously selected action. When all three action dimensions are chosen, the underlying environment progresses to $\boldsymbol{s}_{t+1}$. Equality of Q values is noted where marked with vertical lines.

we must define how the time discounting works. We define the lower MDP to have zero discount for all steps except for when the real environment changes state. Thus, the discount is 0 for all all $\boldsymbol{u}_k^{\boldsymbol{s}_t}$ where $k < N$, and the same as the top MDP when $k = N$. By doing this, the following is then true:

$$Q^U(\boldsymbol{s_t}, \boldsymbol{a_t}) = Q^L(\boldsymbol{u_{N-1}^{s_t}}, a_t^N) \tag{3}$$

where $\boldsymbol{u_{N-1}^{s_t}}$ contains the upper state and the previous $N-1$ actions: $(\boldsymbol{s_t}, \boldsymbol{a_t}^{1:N-1})$. This equality allows us to "short circuit" the backups. Backups are only needed up until the point where the upper MDP can be used improving training and stability.

During training, we parameterize $Q^U$ and $Q^L$ as neural networks. We learn $Q^U$ via TD-0 learning by minimizing:

$$l_{td} = \mathbb{E}_{(\boldsymbol{s_t}, \boldsymbol{a_t}, \boldsymbol{s_{t+1}}) \in R}[(r + \gamma Q^U(\boldsymbol{s_{t+1}}, \pi(\boldsymbol{s_{t+1}})) - Q^U(\boldsymbol{s_t}, \boldsymbol{a_t}))^2]. \tag{4}$$

Next, we learn $Q^L$ by also doing $Q$-learning, but we make use of of the equality noted in equation 3 and zero discounting. There is no new information nor environment dynamics for this MDP. As such we can draw samples from the same replay buffer used for learning $Q^U$. For states $\boldsymbol{u_k^{s_t}}$ where $k < N$ we minimize the bellman error as follows:

$$l_{inner} = \mathbb{E}_{(\boldsymbol{s}, \boldsymbol{a}) \in R} \sum_{k=1}^{N-1} [Q^L(\boldsymbol{u^s}_{k-1}, a^k) - \max_{a^{k+1} \in \mathcal{A}^{k+1}} Q^L(\boldsymbol{u^s}_k, a^{k+1})]^2. \tag{5}$$

When $Q^U$ and $Q^L$ should be equal, as defined in equation 3, we do not backup we instead enforce soft equality by MSE.

$$l_{base} = \mathbb{E}_{(\boldsymbol{s}, \boldsymbol{a}) \in R}[Q^U(\boldsymbol{s}, a) - Q^L((\boldsymbol{s}, \boldsymbol{a}^{1:N-1}), a^N))]^2. \tag{6}$$

In practice, as in DQN, we can also make use of target networks and/or double DQN (Hasselt et al., 2016) when training $Q^U$ and $Q^L$ for increased stability.

When using this model as a policy we compute the argmax over each action dimension of the lower MDP. As with DQN, we employ exploration when training with either epsilon greedy exploration or Boltzmann exploration.

### 2.3 NEURAL NETWORK PARAMETERIZATION

$Q^U$ is a MLP whose inputs are state and actions and outputs are $Q$ values. Unlike in DDPG, the loss function does not need to be smooth with respect to actions. As such, we also feed in discretized representation of each action dimension to make training simpler.

We worked with two parameterizations for $Q^L$. First, we looked at a recurrent LSTM model (Hochreiter & Schmidhuber, 1997). This model has shared weights and passes information via hidden activations from one action dimension to another. The input at each time step is a function of the

current state from the upper MDP, $s_t$, and a single action dimension, $a^i$. As it's an LSTM, the hidden state is capable of accumulating the previous actions. Second, we looked at a version with separate weights for each step of the lower MDP. The lower MDP does not have a fixed size input as the amount of action dimensions it has as inputs varies. To combat this, we use $N$ separate models that are switched between depending on the state index of the lower MDP. We call these distinct models $Q^i$ where $i \in [1, N]$. These models are feed forward neural networks that take as input a concatenation of all previous action selections, $a_t^1 : i$, as well as the upper state, $s_t$. Doing this results in switching $Q^L$ with the respective $Q^i$ in every use. Empirically we found that this weight separation led to more stable training.

In more complex domains, such as vision based control tasks for example, one should untie only a subset of the weights and keep common components – a vision system – the same. In practice, we found that for the simplistic domains we worked in fully untied weights was sufficient. Architecture exploration for these kinds of models is still ongoing work. For full detail of model architectures and training procedures selection see Appendix C.

## 3 RELATED WORK

Our work was motivated by two distinct desires – to learn policies over exponentially large discrete action spaces, and to approximate value functions over high dimensional continuous action spaces effectively. In our paper we used a sequential parameterization of policies that help us to achieve this without making an assumption about the actual functional form of the model. Other prior work attempts to handle high dimensional action spaces by assuming specific decompositions. For example, (Sallans & Hinton, 2004) were able to scale up learning to extremely large action sizes by factoring the action value function and use product of experts to learn policies. An alternative strategy was proposed in (Dulac-Arnold et al., 2015) using action embeddings and applying k-nearest neighbors to reduce scaling of action sizes. By laying out actions on a hypercube, (Pazis & Parr, 2011) are able to perform a binary search over actions resulting in a logarithmic search for the optimal action. Their method is similar to SDQN, as both construct a $Q$-value from sub $Q$-values. Their approach presupposes these constraints, however, and optimizes the Bellman equation by optimizing hyperplanes independently thus enabling optimizing via linear programming. Our approach is iterative and refines the action selection, which contrasts to their independent sub-plane maximization. Pazis & Lagoudakis (2009) and Pazis & Lagoudakis (2011) proposes a transformation similar to ours where a continuous action MDP is converted to a sequence of transitions representing a binary search over the continuous actions. In our setting, we used a 2-layer hierarchy of variable width as opposed to a binary tree. Additionally, we used the original MDP as part of our training procedure to reduce estimation error. We found this to be critical to reduce overestimation error when working with function approximators.

Along with the development of discrete space algorithms, researchers have innovated specialized solutions to learn over continuous state and action environments including (Silver et al., 2014; Lillicrap et al., 2015; Gu et al., 2016b). More recently, novel deep RL approaches have been developed for continuous state and action problems. TRPO (Schulman et al., 2015) and A3C (Mnih et al., 2016) uses a stocastic policy parameterized by diagonal covariance Gaussian distributions. NAF (Gu et al., 2016b) relies on quadratic advantage function enabling closed form optimization of the optimal action. Other methods structure the network in a way such that they are convex in the actions while being non-convex with respect to states (Amos et al., 2016) or use a linear policy (Rajeswaran et al., 2017).

In the context of reinforcement learning, sequential or autoregressive policies have previously been used to describe exponentially large action spaces such as the space of neural architectures, (Zoph & Le, 2016) and over sequences of words (Norouzi et al., 2016; Shen et al., 2015). These approaches rely on policy gradient methods whereas we explore off-policy methods. Hierarchical/options based methods, including (Dayan & Hinton, 1993) which perform spatial abstraction or (Sutton et al., 1999) that perform temporal abstraction pose another way to factor action spaces. These methods refine their action selection from time where our approaches operates on the same timescale and factors the action space.

A vast literature on constructing sequential models to solve tasks exists outside of RL. These models are a natural fit when the data is generated in a sequential process such as in language modeling

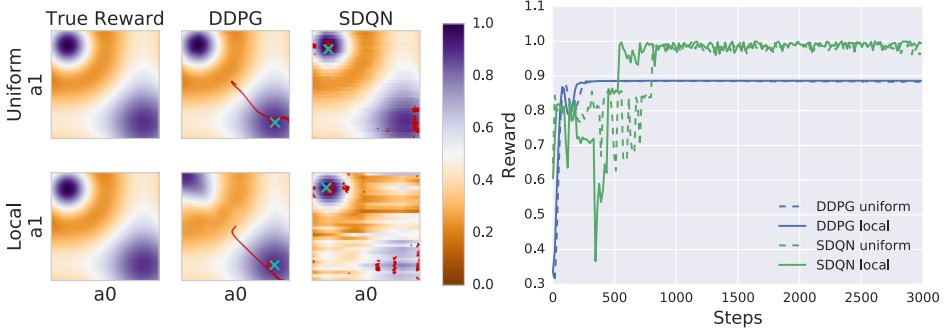

Figure 2: Left: Final reward/$Q$ surface for each algorithm tested. Final policy is marked with a green ×. Policies at previous points in training are denoted with red dots. The SDQN model is capable of performing global search and thus finds the global maximum. The top row contains data collected uniformly over the action space. SDQN and DDPG use this to accurately reconstruct the target $Q$ surface. In the bottom row, actions are sampled from a normal distribution centered on the policy. This results in more sample efficiency but yields poor approximations of the $Q$ surface outside of where the policy is. Right: Reward achieved over time. DDPG quickly converges to a local maximum. SDQN has high variance performance initially as it searches the space, but then quickly converges to the global maximum as the $Q$ surface estimate becomes more accurate.

(Bengio et al., 2003). One of the first and most effective deep learned sequence-to-sequence models for language modeling was proposed in (Sutskever et al., 2014b), which used an encoder-decoder architecture. In other domains, techniques such as NADE (Larochelle & Murray, 2011) have been developed to compute tractable likelihood. Techniques like Pixel RNN (Oord et al., 2016) have been used to great success in the image domain where there is no clear generation sequence. Hierarchical softmax (Morin & Bengio, 2005) performs a hierarchical decomposition based on WordNet semantic information.

The second motivation of our work was to enable learning over more flexible, possibly multimodal policy landscape. Existing methods use stochastic neural networks (Carlos Florensa, 2017) or construct energy models (Haarnoja et al., 2017) sampled with Stein variational gradient descent (Liu & Wang, 2016; Wang & Liu, 2016).

## 4 EXPERIMENTS

### 4.1 MULTIMODAL EXAMPLE ENVIRONMENT

To consider the effectiveness of our algorithm, we consider a deterministic environment with a single time step, and a 2D action space. This can be thought of as being a two-armed bandit problem with deterministic rewards, or as a search problem in 2D action space. We chose our reward function to be a multimodal distribution as shown in the first column in Figure 2. A large suboptimal mode and a smaller optimal mode exist. As with bandit problems, this formulation helps us isolate the ability of our method to find an optimal policy, without the confounding effect that arises from backing up rewards via the Bellman operator for sequential problems.

As in traditional RL, we do exploration while learning. We consider uniformly sampling ($\epsilon$-greedy with $\epsilon = 1$) as well as sampling data from a normal distribution centered at the current policy – we refer to this as "local." A visualization of the final $Q$ surfaces as well as training curves can be found in Figure 2.

DDPG uses local optimization to learn a policy on a constantly changing estimate of $Q$ values predicted by a critic. The form of the $Q$ distribution is flexible and as such there is no closed form properties we can make use of for learning a policy. As such, gradient descent, a local optimization algorithm, is used. This algorithm can get stuck in a sub-optimal policy. We hypothesize that these local maximum in policy space exist in more realistic simulated environments as well. Traditionally, deep learning methods use local optimizers and avoid local minima or maxima by working in a high dimensional parameter space (Choromanska et al., 2015). In RL, however, the action space of a

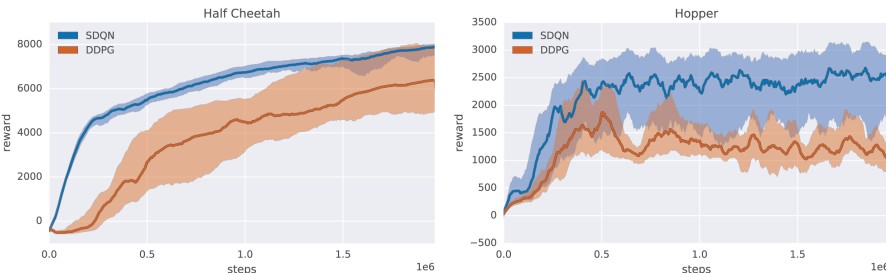

Figure 3: Learning curves of highest performing hyper parameters trained on Mujoco tasks. We show a smoothed median (solid line) with 25 and 75 percentiles range (transparent line) from the 10 random seeds run. SDQN quickly achieves good performance on these tasks.

policy is relatively small dimensional thus it is much more likely that they exist. For example, in the hopper environment, a common failure mode we experienced when training algorithms like DDPG is to learn to balance instead of moving forward and hopping.

We contrast this to SDQN. As expected, this model is capable of completely representing the $Q$ surface (under the limits of discretization). The optimization of the policy is not done locally however enabling convergence to the optimal policy. Much like DDPG, the $Q$ surface learned can be done on uniform, off policy, data. Unlike DDPG, however, the policy will not get stuck in a local maximum. In the uniform behavior policy setting, the model slowly reaches the right solution. [1] With a behavior policy that is closer to being on-policy (such as the stochastic Gaussian greedy policy referred to above), the rate of convergence increases. Much of the error occurs from selecting over estimated actions. When sampling more on policy, the over estimated data points get sampled more frequently resulting in faster training.

### 4.2 MUJOCO ENVIRONMENTS

To evaluate the relative performance of these models we perform a series of experiments on common continuous control tasks. We test the hopper (3-D action space), swimmer (2-D action space), half cheetah (6-D action space), walker2d (6-D action space) and the humanoid environment (17-D action space) from the OpenAI gym suite (Brockman et al., 2016). [2]

We performed a wide hyper parameter search over various parameters in our models (described in Appendix C), and selected the best performing runs. We then ran 10 random seeds of the same hyper parameters to evaluate consistency and to get a more realistic estimate of performance. We believe this replication is necessary as many of these algorithms are not only sensitive to both hyper parameters but random seeds.

First, we look at learning curves of some of the environments tested in Figure 3. Our method quickly achieves good policies much faster than DDPG. For a more qualitative analysis, we use the best reward achieved while training averaged across over 25,000 steps and with evaluations sampled every 5,000 steps. Again we perform an average over 10 different random seeds. This metric gives a much better sense of stability than the traditionally reported instantaneous max reward achieved during training.

We compare our algorithm to the current state-of-the-art in off-policy continuous control: DDPG. Through careful model selection and extensive hyper parameter tuning, we train DDPG agents with performance better than previously published for some of these tasks. Despite this search, however, we believe that there is still space for *significant* performance gain for all the models given different neural network architectures and hyper parameters. See (Henderson et al., 2017; Islam et al., 2017)

---

[1]This assumes that the models have enough capacity. In a limited capacity setting, one would still want to explore locally. Much like SDQN models will shift capacity to modeling the spaces, which are sampled, thus making better use of the capacity.

[2] For technical reasons, our simulations for all experiments use a different numerical simulation strategy provided by Mujoco (Todorov et al., 2012). In practice though, we found the differences in final reward to be within the expected variability of rerunning an algorithm with a different random seed.

| agent | hopper | swimmer | half cheetah | humanoid | walker2d |
|-------|--------|---------|--------------|----------|----------|
| SDQN | **3342.62** | **179.23** | **7774.77** | **3096.71** | 3227.73 |
| DDPG | 3296.49 | 133.52 | 6614.26 | 3055.98 | **3640.93** |

Figure 4: Maximum reward achieved over training averaged over a 25,000 step window with evaluations every 5,000 steps. Results are averaged over 10 randomly initialized trials with fixed hyper parameters. SDQN models perform competitively as compared to DDPG.

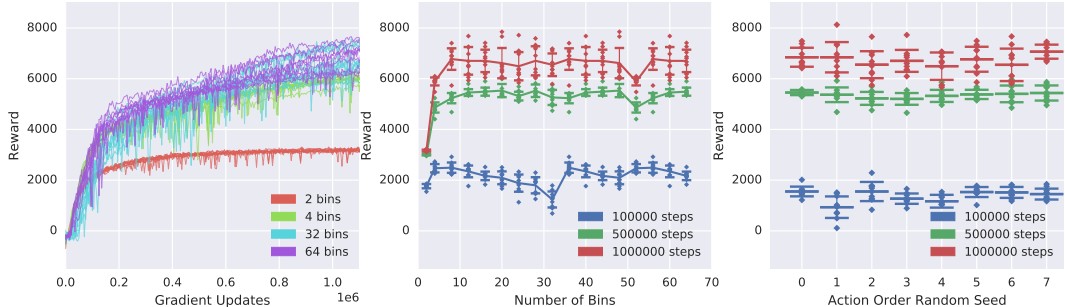

Figure 5: Hyper parameter sensitivity run on Half Cheetah. Left: Learning curves of different numbers of Bins. Center: Comparison of reward versus number of bins evaluated at 3 time points during training. Error bars show 1 std. The number of bins negatively impacts performance for small values of 2 and 4. For values larger than this, however, there is very little change in performance. Right: Comparison of action order for 8 different action orderings evaluated at 3 points during training. Error bars show 1 std. Hyper parameters found above were tuned with the seed=0. In this sample, all orderings achieve similar performance.

for discussion on implementation variability and performance. Results can be seen in Figure 4. Our algorithm achieves better performance on four of the five environments we tested.

### 4.3 EFFECT OF NUMBER OF BINS

Unlike existing continuous control algorithms, we have a choice over the number of discritization bins we choose, $B$. To test the effect of this we first take the best performing half cheetah hyper parameter configuration found above, and rerun it varying the number of bins. For statistical significance we run 10 trials per tested value of $B$. Results can be found in Figure 5. These results suggest that SDQN is robust to this hyper parameter, working well in all bin amounts greater than 4. Lower than 4 bins does not yield enough fine grain enough control to solve this task effectively.

### 4.4 EFFECT OF ACTION ORDER

Next we look to the effect of action order. In most existing environments there is no implicit "ordering" to environments. Given the small action space dimensionality, we hypothesized that this ordering would not matter. We test this hypothesis by taking our hyper parameters for half cheetah found in section 4.2 and reran them with random action ordering, 10 times for statistical significance. Half cheetah has 6 action dimensions thus we are only exploring a subset of orderings. Seed 0 represents the ordering used when finding the original hyper parameters. Results can be found in Figure 5. While there is some variability, the overall changes in performance are small validating our original assumption.

## 5 DISCUSSION

Conceptually, our approach centers on the idea that action selection at each stage can be factored and sequentially selected. In this work we use 1-D action spaces that are discretized as our base component. Existing work in the image modeling domain suggests that using a mixture of logistic units (Salimans et al., 2017) greatly speeds up training and would also satisfy our need for a closed form max. Additionally, this work imposes a prespecified ordering of actions which may negatively

impact training for certain classes of problems (with much larger number of action dimensions). To address this, we could learn to factor the action space into the sequential order for continuous action spaces or learn to group action sets for discrete action spaces. Another promising direction is to combine this approximate max action with gradient based optimization procedure. This would relieve some of the complexity of the modeling task of the maxing network, at the cost of increased compute when sampling from the policy. Finally, the work presented here is exclusively on off-policy methods. We chose to focus on these methods due to their sample efficiency. Use of an sequential policies with discretized actions could also be used as the policy for any stochastic policy optimization algorithm such as TRPO (Schulman et al., 2015) or A3C (Mnih et al., 2016).

## 6 CONCLUSION

In this work we present a continuous control algorithm that utilize discretized action spaces and sequential models. The technique we propose is an off-policy RL algorithm that utilizes sequential prediction and discretization. We decompose our model into a hierarchy of $Q$ function. The effectiveness of our method is demonstrated on illustrative and benchmark tasks, as well as on more complex continuous control tasks.

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

# Appendix

## A   MODEL DIAGRAMS

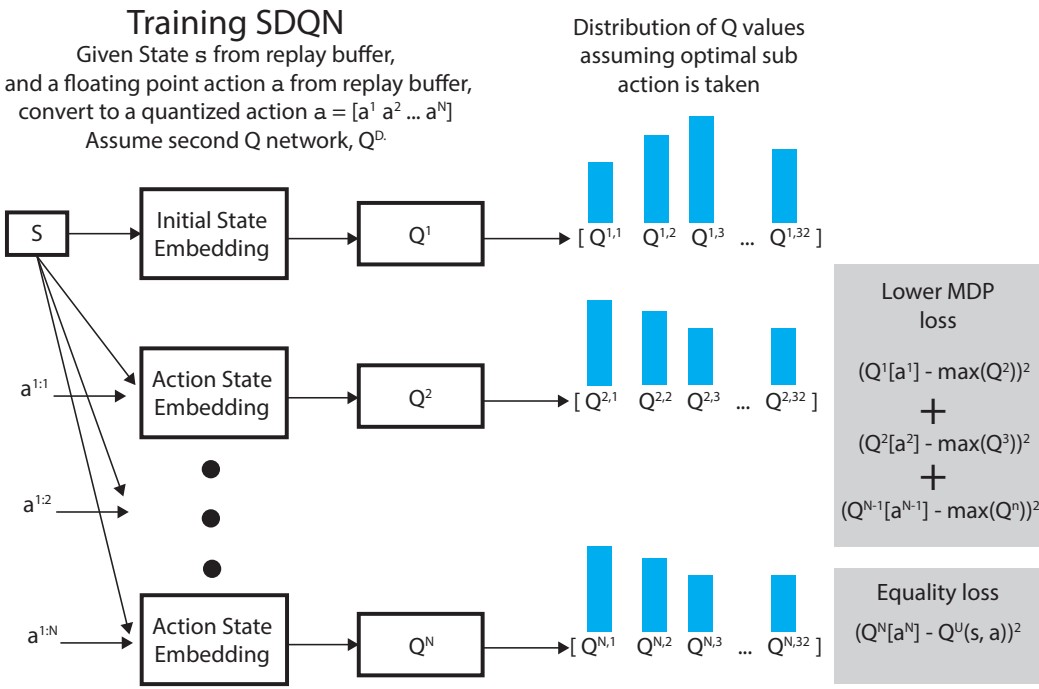

Figure A.1: Pictorial view for the SDQN network showing training. In this figure we train the entire lower MDP, $Q^L$. $Q^L$ is made up of $Q^i$ where $i \in [1, N]$. See Figure A.2 for model in evaluation mode.

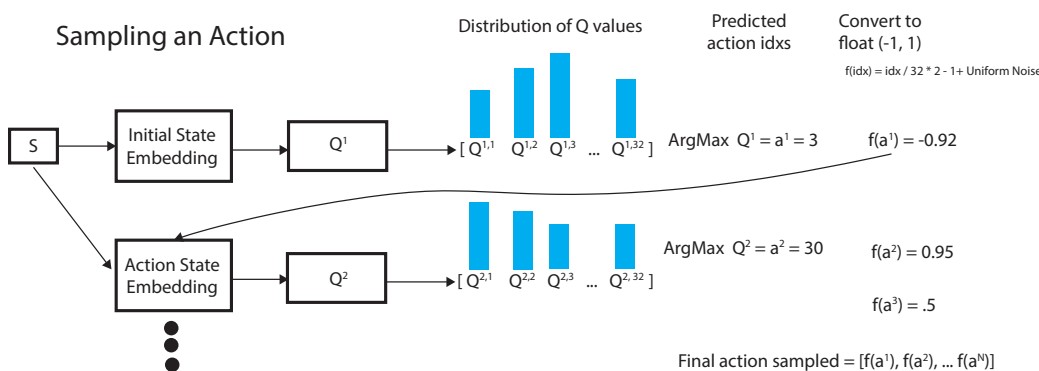

Figure A.2: Pictorial view of sampling actions with SDQN. Each action dimension is computed by taking an argmax of each $Q^i$ for $i \in [1, N]$.

## B   MODEL VISUALIZATION

To gain insight into the characteristics of $Q$ that our SDQN algorithm learns, we visualized results from the hopper environment as it is complex but has a small dimensional action space.

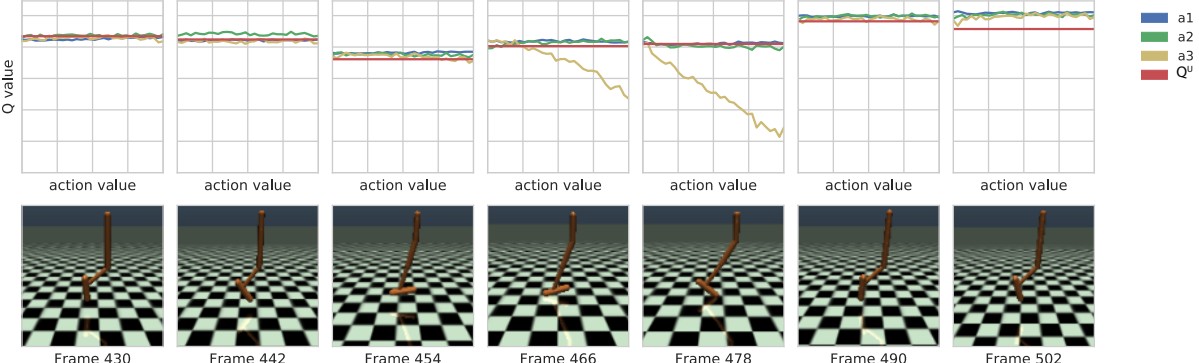

Figure B.3: Exploration of the sub-DQN during after training. The top row shows the $Q^i$ predictions for a given frame (action dimensions correspond to the joint starting at the top and moving toward the bottom – action 3 is the ankle joint). The bottom row shows the corresponding rendering of the current state. For insensitive parts of the gait, such as when the hopper is in the air (e.g. frame 430, 442, 490, 502), the network learns to be agnostic to the choice of actions; this is reflected in the flat Q-value distribution, viewed as a function of action index. On the other hand, for critical parts of the gait, such as when the hopper is in contact with the ground (e.g. frames 446, 478), the network learns that certain actions are much better than others, and the Q-distribution is no longer a flat line. This reflects the fact that taking wrong actions in these regimes could lead to bad results such as tripping, yielding a lower reward.

First we compute each action dimension's $Q$ distribution, $Q^L$ / $Q^i$, and compare those distributions to that of the top MDP for the full action dimentionality, $Q^U$. A figure containing these distributions and corresponding state visualization can be found in Figure B.3.

For most states in the hopper walk cycle, the $Q$ distribution is very flat. This implies that small changes in the action taken in a state will have little impact on future reward. This makes sense as the system can recover from any action taken in this frame. However, this is not true for all states – certain critical states exist, such as when the hopper is pushing off, where not selecting the correct action value greatly degrades performance. This can be seen in frame 466.

Our algorithm is trained with a number of soft constraints. First, if fully converged, we would expect $Q^{i-1} >= Q^i$ as every new sub-action taken should maintain or improve the expected future discounted reward. Additionally, we would expect $Q^N(s,a) = Q^U(s,a)$ (from equation 6). In the majority of frames these properties seem correct, but there is certainly room for improvement.

Next, we attempt to look at $Q$ surfaces in a more global manner. We plot 2D cross sections for each pair of actions and assume the third dimension is zero. Figure B.4 shows the results.

As seen in the previous visualization, the surface of both the sequential $Q$ surface and the $Q^U$ is not smooth, which is expected as the environment action space for Hopper is expected to be highly non-linear. Some regions of the surface seem quite noisy which is not expected. Interestingly though, these regions of noise do not seem to lower the performance of the final policy. In $Q$-learning, only the maximum $Q$ value regions have any impact on the taken policy. Future work is needed to better characterize this effect. We would like to explore techniques that use "soft" Q-learning Nachum et al. (2017); Schulman et al. (2017); Haarnoja et al. (2017). These techniques will use more of the $Q$ surface thus smooth the representations.

Additionally, we notice that the dimensions of the autoregressive model are modeled differently. The last action, $a_3$ has considerably more noise than the previous two action dimensions. This large difference in the smoothness and shape of the surfaces demonstrates that the order of the actions dimensions matters. This figure suggests that the model has a harder time learning sharp features

in the $a_1$ dimension. In future work, we would like to explore learned orderings, or bidirectional models, to combat this.

Finally, the form of $Q^U$ is extremely noisy and has many cube artifacts. The input of this function is both a one hot quantized action, as well as the floating point representation. It appears the model uses the quantization as its main feature and learns a sharp $Q$ surface.

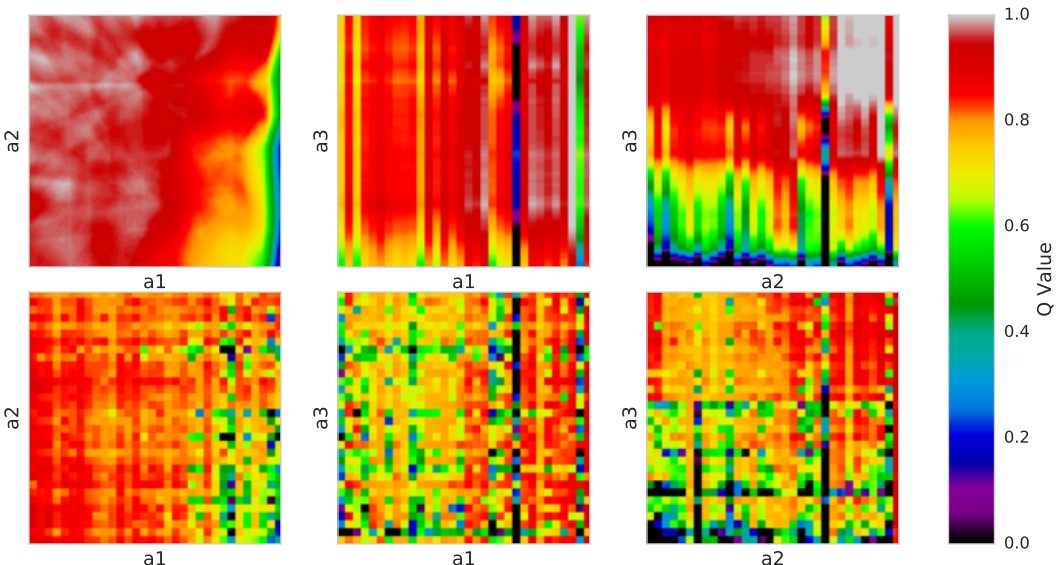

Figure B.4: $Q$ surfaces given a fixed state. Top row is the autoregressive model, $Q^N$. The bottom row is the double DQN, $Q^U$. We observe high noise in both models. Additionally, we see smoother variation in earlier action dimensions, which suggests that order may matter when in limited capacity regimes. $Q$ values are computed with a reward scale of 0.1, and a discounted return of 0.995.

## C  TRAINING AND MODEL DETAILS

### C.1  HYPER PARAMETER SENSITIVITY

The most sensitive hyper parameters were the learning rate of the two networks, reward scale, and finally, discount factor. Parameters such as batch size, quantization amounts, and network sizes mattered to a lesser extent. We found it best to have exploration be done on less than 10% of the transitions. We didn't see any particular value below this that gave better performance. In our experiments, the structure of model used also played a large impact in performance, such as, using tied versus untied weights for each sequential prediction model.

In future work, we would like to lower the amount of hyper parameters needed in these algorithms and study the effects of various hyper parameters more thoroughly.

### C.2  SDQN

In this model, we looked at a number of configurations. Of the greatest importance is the form of the model itself. We looked at an LSTM parameterization as well as an untied weight variant. The untied weight variant's parameter search is listed below.

To compute $Q^i$ we first take the state and previous actions and do one fully connected layer of size "embedding size". We then concatenate these representations and feed them into a 2 hidden

layer MLP with "hidden size" units. The output of this network is "quantization bins" wide with no activation.

$Q^U$ uses the same embedding of state and action and then passes it though a 1 hidden layer fully connected network finally outputting a single scalar.

| Hyper Parameter | Range | Notes |
|---|---|---|
| use batchnorm | on, off | use batchnorm on the networks |
| replay capacity: | 2e4, 2e5, inf | |
| batch size | 256, 512 | |
| quantization bins | 32 | We found higher values generally converged to better final solutions. |
| hidden size | 256, 512 | |
| embedding size | 128 | |
| reward scaling | 0.05, 0.1 | |
| target network moving average | 0.99, 0.99, 0.98 | |
| adam learning rate for TD updates | 1e-3, 1e-4, 1e-5 | |
| adam learning rate for maxing network | 1e-3, 1e-4, 1e-5 | |
| gradient clipping | off, 10 | |
| l2 reguralization | off, 1e-1, 1e-2, 1e-3, 1e-4 | |
| learning rate decay | log linear, none | |
| learning rate decay delta | -2 | Decay 2 orders of magnitude down from 0 to 1m steps. |
| td term multiplier | 0.2, 0.5, | |
| using target network on double q network | on, off | |
| tree consistency multiplier | 5 | Scalar on the tree loss |
| energy use penalty | 0, 0.05, 0.1, 0.2 | Factor multiplied by velocity and subtracted from reward |
| gamma (discount factor) | 0.9, 0.99, 0.999 | |
| drag down reguralizer | 0.0, 0.1 | Constant factor to penalize high q values. This is used to control over exploration. It has a very small effect in final performance. |
| tree target greedy penalty | 1.0 | A penalty on MSE or Q predictions from greedy net to target. This acts to prevent taking too large steps in function space |
| exploration type | boltzmann or epsilon greedy | |
| boltzman temperature | 1.0, 0.5, 0.1, 0.01, 0.001 | |
| prob to sample from boltzman (vs take max) | 1.0, 0.5, 0.2, 0.1, 0.0 | |
| boltzman decay | decay both prob to sample and boltzman temperature to 0.001 | |
| epsilon noise | 0.5, 0.2, 0.1, 0.05, 0.01 | |
| epsilon decay | linearly to 0.001 over the first 1M steps | |

Best hyper parameters for a subset of environments.

| Hyper Parameter | Hopper | Cheetah |
|---|---|---|
| use batchnorm | off | off |

| | | |
|---|---|---|
| replay capacity: | inf | inf |
| batch size | 512 | 512 |
| quantization bins | 32 | 32 |
| hidden size | 256 | 512 |
| embedding size | 128 | 128 |
| reward scaling | 0.1 | 0.1 |
| target network moving average | 0.99 | 0.9 |
| adam learning rate for TD updates | 1e-3 | 1e-3 |
| adam learning rate for maxing network | 5e-5 | 1e-4 |
| gradient clipping | off | off |
| l2 regularization | 1e-4 | 1e-4 |
| learning rate decay for q | log linear | log linear |
| learning rate decay delta for q | 2 orders of magnitude down from interpolated over 1M steps. | 2 orders down interpolated over 1M steps |
| learning rate decay for tree | none | none |
| learning rate decay delta for tree | NA | NA |
| td term multiplier | 0.5 | 0.5 |
| useing target network on double q network | off | on |
| tree consistency multiplier | 5 | 5 |
| energy use penalty | 0 | 0.0 |
| gamma (discount factor) | 0.995 | 0.99 |
| drag down reguralizer | 0.1 | 0.1 |
| tree target greedy penalty | 1.0 | 1.0 |
| exploration type | boltzmann | boltzmann |
| boltzman temperature | 1.0 | 0.1 |
| prob to sample from boltzman (vs take max) | 0.2 | 1.0 |
| boltzman decay | decay both prob to sample and boltzman temperature to 0.001 over 1M steps | decay both prob to sample and boltzman temperature to 0.001 over 1M steps |
| epsilon noise | NA | NA |
| epsilon decay | NA | NA |

## D    APPENDIX PSEUDOCODE

---

**Algorithm 1** SDQN sampling from policy

---

1: Assuming parameter's $\phi$ are initialized.
2: **procedure** $\pi(s)$                                                               ▷ Sample from the $Q$ values
3:     $\mathbf{a} \leftarrow []$
4:     **for** i...N **do**
5:         $a^i_{bin} = \text{argmax}_{a_i} Q^i(s, a)$                          ▷ Find the max bin idx
6:         $a^i = a^i_{bin}/B + rand()/B$ ▷ Convert the integer bin into a continuous value randomly in that bin. Assume $B$ bins.
7:         **if** $\epsilon >$ rand() **then**                                      ▷ Epsilon Greedy exploration
8:             $a^i \leftarrow rand()$
9:         Append $a^i$ to $\mathbf{a}$: $\mathbf{a} \leftarrow [\mathbf{a}; \mathbf{a^i}]$
10:     **return** $a$

---

---

**Algorithm 2** SDQN training

---

1: **Initialization**
2: Initialize replay buffer, $R$ to be empty.
3: Initialize the environment.
4: Randomly initialize $\theta, \theta_{target}, \phi$
5: **for** i...1000 **do**                                          ▷ Add initial data to $R$
6:     $s_e \leftarrow$ Current environment state
7:     $a_e \leftarrow \pi(s_e; \phi)$                                                 ▷ see 1
8:     Execute $a_e$ in the environment receiving $r_e$, and $s_{e+1}$
9:     Add transition $(s_e, a_e, r_e, s_{e+1})$ to replay buffer $R$
10:     If the environment is finished, reset it.

11: **while** Training **do**
12:     **Policy and critic update**
13:     Sample a batch of data, $(s_t, a_t, r_t, s_{t+1})$ from $R$
14:     $y_{td} = r_t + Q^U(s_t, \pi(s_t; \phi); \theta_{target})$                       ▷ Equation 4
15:     $l_{td} = y_{td} - Q^U(s_t, a_t; \theta)$                               ▷ Equation 4
16:     $l_{base} = [Q^U(s_t, a_t; \theta) - Q^N((s_t, a_t^{1:N-1}), a_t^N; \phi)]^2$     ▷ Equation 6
17:     $y_{inner} = \max_{a^{i+1} \in \mathcal{A}^{i+1}} Q^{i+1}(\boldsymbol{s}, [\boldsymbol{a}^{1:i}, a^{i+1}]; \theta)$      ▷ Equation 5
18:     $l_{inner} = \frac{1}{N-1} \sum_{i=0}^{N-1} [Q^i(\boldsymbol{s}, \boldsymbol{a}^{1:i}; \theta) - y_{inner}]^2$      ▷ Equation 5
19:     Update $\theta$ using Adam with $\dfrac{dl_{td}}{d\theta}$
20:     Update $\phi$ using Adam with $\dfrac{d(l_{inner} + l_{base})}{d\phi}$ assuming $\dfrac{dy_{inner}}{d\phi} = 0$
21:     Update $\theta_{target} \leftarrow \theta_{target}\text{decay} + \theta(1 - \text{decay})$

22:     **Add a transition to $R$**
23:     $s_e \leftarrow$ Current environment state
24:     $a_e \leftarrow \pi(s_e; \phi)$
25:     Execute $a_e$ in the environment receiving $r_e$, and $s_{e+1}$
26:     Add transition $(s_e, a_e, r_e, s_{e+1})$ to $R$
27:     If the environment is finished, reset it.

---

