# OpenReview forum: "Discrete Sequential Prediction of Continuous Actions for Deep RL"
_ICLR.cc/2018/Conference — Reject_

### Official Review · AnonReviewer3 · 2017-11-15
**The paper presents a correct, reasonably effective, but not groundbreaking approach to Q value approximation in MDPs with high-dimensional action spaces**

**Rating:** 7
**Confidence:** 5

**Review:**

The paper presents Sequential Deep Q-Networks (SDQNs), which select actions from discretized high-dimensional action spaces.  This is done by introducing another, undiscounted MDP in which each action dimension is chosen sequentially by an agent.  By training a Q network to best choose these action dimensions, and loosely enforcing equality between the original and new MDPs at points where they are equivalent, the new MDP can be successfully navigated, resulting in good action selection for the original MDP.  This is experimentally compared against DDPG in several domains.  There are no theoretical results.

This work is correct and clearly written.  Experiments do demonstrate improved effectiveness in the chosen domains, and the authors do a nice job of illustrating the range of performance by their approach (which has low variance in some domains, but high variance in others).  Because of the clarity of the paper, the effectiveness of the approach, and the high quality experiments, I encourage acceptance.

It doesn't strike me as world-changing, however.  The MDP-within-an-MDP approach is quite similar to the Pazis and Lagoudakis MDP decomposition for the same problem (work which is appropriately cited, but maybe too briefly compared against).  In other words, it strikes me as merely being P&L plus networks, dampening my enthusiasm.

My one question for the authors is how much the order of action dimension selection matters.  This seems probably quite important practically, but is undiscussed.

---

> ### Author Response · Authors · 2018-01-03
> **rebutal**
>
> Thank you for your thoughtful review.
>
> We were not actually aware of Pazis and Lagoudakis's work on this subject (we did cite them, but for another one of there papers, not the paper we believe you are referencing.). We have updated the text to include a section on this work. As per the differences, we are using neural network function approximators. Naively applying this decomposition increases the time dependences in the MDP, as such when using function approximators error accumulates. While attempting to train like this does work, it is incredibly unstable and hyperparameter sensitive. Our second contribution is thus a modified way of training these networks by training the hierarchy of MDP together -- using the upper to bootstrap the lower. This, unlike the original PL-like algorithm, is much more stable as it reduces function overestimation approximator error. With these improvements we are able to train on more complex tasks than originally explored.
>
> As per your question on action ordering: we have an experiment (section 4.4). We found on that problem at least that there was little to no change in performance given different action orders.

---

### Official Review · AnonReviewer1 · 2017-11-28
**A useful method to handle continuous action spaces; however comes with additional cost of training as many networks as the number of actions**

**Rating:** 5
**Confidence:** 1

**Review:**

Originality
--------------
When the action space is N-dimensional, computing argmax could be problematic. The paper proposes to address the problem by creating N MDPs with 1-D actions.

Clarity
---------
1) Explicitly writing down DDPG will be helpful
2) The number of actions in each of the domains will also be useful

Quality
----------
1) The paper reports experimental results on order of actions as well as binning, and the results confirm with what one would expect from intuition.
2) It will be important to talk about the case when the action dimension N is very large, what happens in that case? Does the proposed method would work in such a scenario? A discussion is needed.
3) Given that the ordering of actions does not matter, what is the real take away of looking at them as 'sequence' (which has not temporal structure because action order could be arbitrary)?


Significance
----------------
While the proposed method seems a reasonable approach to handle the argmax problem, it still requires training multiple networks for Q^i (i=1,..N) for Q^L, which is a limitation. Further, since the actions could be arbitrary, it is unclear where 'sequence' approach helps. These limit the understand and hence significance.

---

> ### Author Response · Authors · 2018-01-03
> **rebuttal**
>
> Thank you for your thoughtful review. We will try to address your concerns, as follows:
>
> # Clarity
> We agreed on both fronts, and we have updated the text.
>
> # Quality
> 2. Huge action dimensions would be an interesting application but are outside the scope of our focus: continuous control for robotics tasks. Theoretically, our algorithm scales linearly in terms of compute but exponentially in terms of the MDP (although in practice there is a lot of independence between actions which again makes it closer to linear scaling). We would expect that as N grows, learning the lower MDP will become harder and harder due to the increased temporal dependencies. For very large N, we would almost surely expect that one would need a logarithmic hierarchy or a technique similar to [1].
> 3. As a baseline while developing this work (not included in this paper), we used algorithms that were not sequence based. This algorithm predicted Q values independently for each action dimension. While the algorithm worked, the lack of action-to-action conditioning greatly restricted the functional form of our model and resulted in sub-par performance.
> The sequence version allows these previously missing action-to-action interactions while keeping maxation tractable. By putting action dimensions in a sequence, we are able to easily condition results on the previous action dimensions. The fact that ordering does not matter is a good thing for us and allows this technique to work! It is possible to construct some set based interaction scheme that has a similar ability to preserve conditioning, but we are not aware of any such constructions that support explicit maxation while retaining this interaction action conditioning.
>
> # Significance
> We do not see training multiple networks as a limitation as long as sample complexity does not suffer (as we have shown in regard to DDPG). In the robotics settings, the compute cost is often much, much smaller in comparison to the hardware / robot cost. The run time is no different than say running a single RNN over the action dimensions, and in terms of memory, these models are also quite small ~ order 0.1 - 1Mb per action dimension depending on network sizes. Additionally, not all of the components need to be separate. In a tasks involving vision, for example, one could use a common feature extractor.
>
> [1]Dulac-Arnold, Gabriel, et al. "Deep reinforcement learning in large discrete action spaces." arXiv preprint arXiv:1512.07679(2015).

---

### Official Review · AnonReviewer2 · 2017-11-28

**Rating:** 4
**Confidence:** 5

**Review:**

The paper describes a new RL technique for high dimensional action spaces.  It discretizes each dimension of the action space, but to avoid an exponential blowup, it selects the action for each dimension in sequence.  This is an interesting approach.  The paper reformulates the MDP with a high dimensional action space into an equivalent MDP with more time steps (one per dimension) that each selects the action in one dimension.  This makes sense.

While I do like very much the model, I am perplex about the training technique.  The lower MDP is precisely the new proposed model with unidimensional actions and therefore it should be sufficient.  However, the paper also describes an upper MDP that seems to be superfluous.  The two MDPs are mathematically equivalent, but their Q-values are obtained differently (TD-0 for the upper MDP and Q-learning for the lower MDP) and yet the paper tries to minimize the Euclidean distance between them.  This is really puzzling since the different training algorithms suggest that the Q-values should be different while minimizing the Euclidean distance between them tries to make them equal.  The paper suggests that divergence occurs without the upper MDP.  This is really suspicious. The approach feels like a band-aid solution to cover a problem that the authors could not identify.  While the empirical results are good, I don't think the paper should be published until the authors figure out a principled way of training.

The proposed approach reformulates the MDP with high dimensional actions into an equivalent one with uni dimensional actions.  There is a catch.  This approach effectively hides the exponential action space into the state space which becomes exponential.  Since u contains all the actions of the previous dimensions, we are effectively increasing the state space by an exponential factor.  The paper should discuss this and explain what are the consequences in practice.  In the end, the MDP does not become simpler.

Overall, this is an interesting paper with a good idea, but the training technique is not mature enough for publication.

---

> ### Author Response · Authors · 2018-01-03
> **rebuttal**
>
> Thank you for your thoughtful review. We will try to address your concerns bellow:
>
> # Hierarchy
> We agree that, in theory, the lower MDP should be sufficient. In practice, as we pointed out in the paper, Q learning with (deep) function approximators is unstable and sensitive to hyperparameters. To our knowledge, this phenomenon is not thoroughly understood. There have been many papers, however, describing potential failure points, proposing a solution, and showing improvement however -- examples include Double DQN [1], Dueling networks [2], all improvements from rainbow networks [3], and many more. We see our 2-layer training in a similar vein to these works. In particular, the training procedure described here is closely related to Double DQN in theory and implementation.
> The issue we seek to address is the failure in the "Bellman backup" through time due to repeated function approximator error. When working with long MDP, learning associations between states and action sequences have been shown to be hard [4]. [4] shows that this effect is so impactful that by lowering the control frequency (increasing the frame-skip count) actually increased performance in some tasks. Additionally, in the policy gradient algorithms, increasing the frequency of states has been shown to increase gradient variance with the number of timesteps [5].
> Initially we explored just the lower MDP and achieved reasonable performance, but the resulting algorithm was incredibly sensitive to hyperparameter and quite unstable, partially due to Q value overestimation. Our hierarchy is a way to address this instability. It does so in a similar manner to that employed in Double DQN; the use of two networks to combat overestimation. Still, solving the root instability of Q-learning with function approximators is an open question and something that interests us greatly.
>
> # Exponential problems
> Thank you for this observation. This is true and it is worth calling more attention to it, which we have done in the text (now updated). The exponential action space does turn into a exponential MDP. Luckily for us though, many problems do not actually require full search of this exponential space. Early in this work, we hypothesized that the space was largely independent between action dimensions. This fact is often exploited in policy gradient approaches as the policy distribution is often parameterized as a diagonal covariance normal[6, 7]. We tested this independence hypothesis in the Q learning settings (using a novel Q learning algorithm, not included in this paper) where the Q values were the sum of terms computed from each action dimension independently) and found that we were able achieve reasonable performance, though not state of the art. In general, we don't expect to be able to search the full exponential space. Early in training the interactions will mostly be linear / independent due to the nature of these neural networks at initialization. As training progresses, we do expect to be able to capture some interaction relationships. In our experiments, adding in this conditioning does increase performance of the final algorithm.
>
>
> [1] Van Hasselt, Hado, Arthur Guez, and David Silver. "Deep Reinforcement Learning with Double Q-Learning." AAAI. 2016.
> [2] Wang, Ziyu, et al. "Dueling network architectures for deep reinforcement learning." arXiv preprint arXiv:1511.06581(2015).
> [3] Hessel, Matteo, et al. "Rainbow: Combining Improvements in Deep Reinforcement Learning." arXiv preprint arXiv:1710.02298 (2017).
> [4] Braylan, Alex, et al. "Frame skip is a powerful parameter for learning to play atari." Space 1600 (2000): 1800.
> [5] Salimans, Tim, et al. "Evolution strategies as a scalable alternative to reinforcement learning." arXiv preprint arXiv:1703.03864 (2017).
> [6]Schulman, John, et al. "Trust region policy optimization." Proceedings of the 32nd International Conference on Machine Learning (ICML-15). 2015.
> [7]Mnih, Volodymyr, et al. "Asynchronous methods for deep reinforcement learning." International Conference on Machine Learning. 2016.

---

### Author Response · Authors · 2018-01-03
**paper update**

We have updated the paper with the following:
- added few sentences on the complexity of action space being shifted into the MDP
- added equation for DDPG update
- added more related work from Pazis and Lagoudakis
- added action space dimensionality for each environment in experiments

---

### Decision · Program_Chairs · 2018-01-29
**ICLR 2018 Conference Acceptance Decision**

**Decision:**

Reject

**Comment:**

The reviewers consider the paper to promising, but raise issues with the increase in the complexity of the MDP caused by the authors' parameterization of the action space, and comparisons with earlier work (Pazis and Lagoudakis).   While the authors cite this work, and say that they that they needed to make changes to PL to make it work in their setting (in addition to adding the deep networks), they do not explicitly show comparisons in the paper to any other discretization schemes.